# Poliovirus excretion among children with primary immune deficiency in Pakistan: a pilot surveillance study protocol

Asma Sadruddin Pethani ![ORCID],[1] Zaubina Kazi,[1] Ujala Nayyar,[2] Muhammad Shafiq-ur-Rehman,[2] Muhammad Tahir Yousafzai,[1] Mach Ondrej,[3] Ali Faisal Saleem ![ORCID] [1]

[1]Pediatrics and Child Health, Aga Khan University, Karachi, Pakistan
[2]Polio Eradication Initiative, World Health Organization Country Office for Pakistan, Islamabad, Pakistan
[3]Polio Department, World Health Organization, Geneva, Switzerland

**Correspondence to**
Dr Ali Faisal Saleem;
ali.saleem@aku.edu

## ABSTRACT

**Introduction** Children with primary immunodeficiency disorders (PID) are more susceptible to developing viral infections and are at a substantially increased risk of developing paralytic poliomyelitis. Such children, if given oral polio vaccines tend to excrete poliovirus chronically that may lead to the propagation of highly divergent vaccine-derived poliovirus (VDPV). Consequently, they may act as a reservoir for the community by introducing an altered virus potentially imposing a risk to global polio eradication. However, the risks of chronic and prolonged excretion are not well characterised in the study context. This study seeks to establish a pilot surveillance system for successful identification and monitoring of VDPV excretion among children with PID. It will assess whether the Jeffrey Modell warning signs of PID can be used as an appropriate screening tool for PID in Pakistan.
**Methods and analysis** In this pilot surveillance, recruitment of PID cases is currently done at participating hospitals in Pakistan. Potential children are screened and tested against the Jeffrey Modell Foundation (JMF) warning signs for immunodeficiency and their stool is collected to test for poliovirus excretion. Cases excreting poliovirus are followed until the two consecutive negative stool samples are obtained over a period of 6 months. The data will be analysed to calculate hospital-based proportions of total Immunodeficiency-related vaccine-derived poliovirus (iVDPV) cases over a 2-year period and to determine the sensitivity and specificity of the JMF signs.
**Ethics and dissemination** This protocol was reviewed and approved by the WHO (WHO Reference-2018/811124-0), Aga Khan University (AKU ERC-2018-0380-1029) and National Bioethics Committee (Ref No. 4-87 NBC-308-Y2). The results will be published in an open access peer-reviewed scientific journal and presented to the iVDPV Working Group members, policy-makers, paediatric consultants and fellow researchers with the same domain interest. It may be presented in scientific conferences and seminars in the form of oral or poster presentations.

## INTRODUCTION

Polio is a highly infectious viral disease which invades the nervous system of a child and

### Strengths and limitations of this study

► Primary immunodeficiency disorders (PID) patients are screened for excretion of poliovirus for the first time in Pakistan.
► Jeffrey Modell warning signs for PID are used together with immunological diagnostic testing for PID.
► Poliovirus excretors are followed up in monthly intervals until two consecutive samples are negative.
► Proportions and types of poliovirus excretors among PID patients are analysed monthly.
► While attempts were made to cover entire Pakistan, the screening may miss PID cases in more remote areas.

may cause irreversible paralysis.[1] The use of the live oral poliovirus vaccine (OPV) in routine and supplementary immunisation activities has significantly reduced the incidence of paralytic poliomyelitis and transmission of wild polioviruses around the globe. This success can be attributed to the ability of this vaccine to induce mucosal immunity as well as serum neutralising antibodies protective against paralytic poliomyelitis.[2] However, individuals with primary immunodeficiency disorders (PID) exposed to OPV through vaccination or household contacts are not only at increased risk of vaccine-associated paralytic poliomyelitis but may also have prolonged excretion of Sabin vaccine-derived polioviruses (VDPVs).[3] Sabin polioviruses are known to have increased transmissibility and neurovirulence following prolonged intestinal replication. Such individuals could theoretically transmit the virus to contacts and the general population following the interruption of wild poliovirus transmission.[4] These polioviruses are called immunodeficiency-related vaccine-derived poliovirus (iVDPVs),

which indicates that these viruses have replicated and evolved in immunocompromised individuals. Previous studies in Sri Lanka, Egypt, Bangladesh and elsewhere demonstrated that it is essential to identify PID persons excreting polioviruses.[1 5 6]

Poliovirus excretion into the environment negates global poliovirus eradication efforts. PID individuals infected with poliovirus are at risk of developing paralysis. Antiviral therapy will be offered to those identified immunodeficient children who are excreting polioviruses to prevent the spread of these polioviruses.

## Aim

Identification and diagnosis of PID is complex and requires specialist interventions. The Jeffrey Modell Foundation (JMF) has developed 10 warning signs for the screening of suspected primary immunodeficiency patients.[7] Although this screening tool has been used in various countries, its sensitivity and specificity in low-resource countries such as Pakistan have not been assessed. The primary aim of this project is to develop a pilot surveillance system for the identification of primary immunodeficient children in sentinel sites of Pakistan which will provide the basis for the development of longer-term surveillance structure for the successful identification, genetic characterisation and monitoring of VDPV excretion. This will help in identifying long-term poliovirus excretors among them, which will in return help in the prevention of future transmission and VDPV outbreaks.

Secondary aims will be to calculate the hospital-based proportion of iVDPV cases and to document the identified iVDPV cases in the WHO global registry list. Finally, it will help assess whether the Jeffrey Modell warning signs of PID are an appropriate screening tool for PID in Pakistan.

## METHODOLOGY
### Project structure

The surveillance of iVDPV is currently being undertaken at participatory tertiary healthcare facilities around Pakistan. This pilot surveillance will help in structuring a sustainable, long-term surveillance for PID children in Pakistan. In the initial stage, mapping was executed to identify participatory hospitals and their consultant paediatricians. Master training workshops were then conducted for these paediatricians from the selected hospitals of each province of Pakistan who agreed to participate in iVDPV surveillance. The project was rolled out as follows:
▶ First 6 months: Aga Khan University (AKU) Hospital+three other major tertiary care hospitals in Karachi
▶ Months 7–24: AKU Hospital+major hospitals in Karachi+tertiary hospitals located in the provinces of Sindh (Hyderabad), Punjab (Lahore, Multan, Islamabad), Balochistan (Quetta), Khyber Pakhtunkhwa and Gilgit-Baltistan.

### Training of master trainers workshop

The master training workshops were conducted all over the country in collaboration with WHO, Pakistan. All tertiary care hospitals in each province with paediatricians were approached. Across Pakistan, 49 tertiary care hospitals have been participating in iVDPV surveillance. The participatory rate from the province Sindh (n=7) and Balochistan (n=3) is 100%, followed by 72.5% (n=29) from the province Punjab. The participatory proportion from the province Khyber Pakhtunkhwa (n=10) is 38.5%, which is the lowest among the remaining provinces (figure 1). The purpose of the training workshop was to train the master trainers who in turn guided their departmental doctors for screening and identification of PID children using the JMF warning signs.

The master training workshops were conducted by the principal investigator (PI) of this project who is a paediatric infectious diseases consultant at the Aga Khan University Hospital in collaboration with WHO experts in Polio Surveillance.

### Coordination process

This study requires close collaboration and coordination among the stakeholders. The Aga Khan University Hospital (Pakistan) is the main coordination centre for the study. A research coordinator (RC) serves as a liaison between the participatory hospitals in each province and the WHO provincial team, which collects the stool samples of identified PID children. National Institute of Health, Islamabad virology lab processes the stool samples and identifies the type of poliovirus.

### Data collection process

Children are identified by the paediatricians trained on Jeffrey Model Foundation warning signs and the diagnosis of primary immunodeficiency. Any child (up to 18 years of age) fulfilling the criteria of two or more positive JMF signs, will be further assessed and tested for HIV. The children who test negative for HIV, either suspected or confirmed PIDs are considered potential iVDPV excreters and are enrolled in the study. Enrolled children will be further tested for complete blood count (CBC), flow cytometry and quantitative measurement of immunoglobulins (QIGs) level after consulting with the coordination cell at the Aga Khan University. A consultant haematologist at the Aga Khan University will review the flow cytometry and immunoglobulins level reports. After enrolment, WHO provincial teams collect two consecutive stool samples, which are examined for poliovirus at National Institute of Health (NIH) Islamabad-Pakistan using standard poliovirus detection methodology used in the Global Polio Laboratory Network. The second stool sample will be collected 24 hours after the first sample. Any child found to be positive will be followed up with monthly stool samples for 6 months or till two subsequent samples become negative. All iVDPVs detected will be documented in the WHO global registry list of known iVDPV cases.

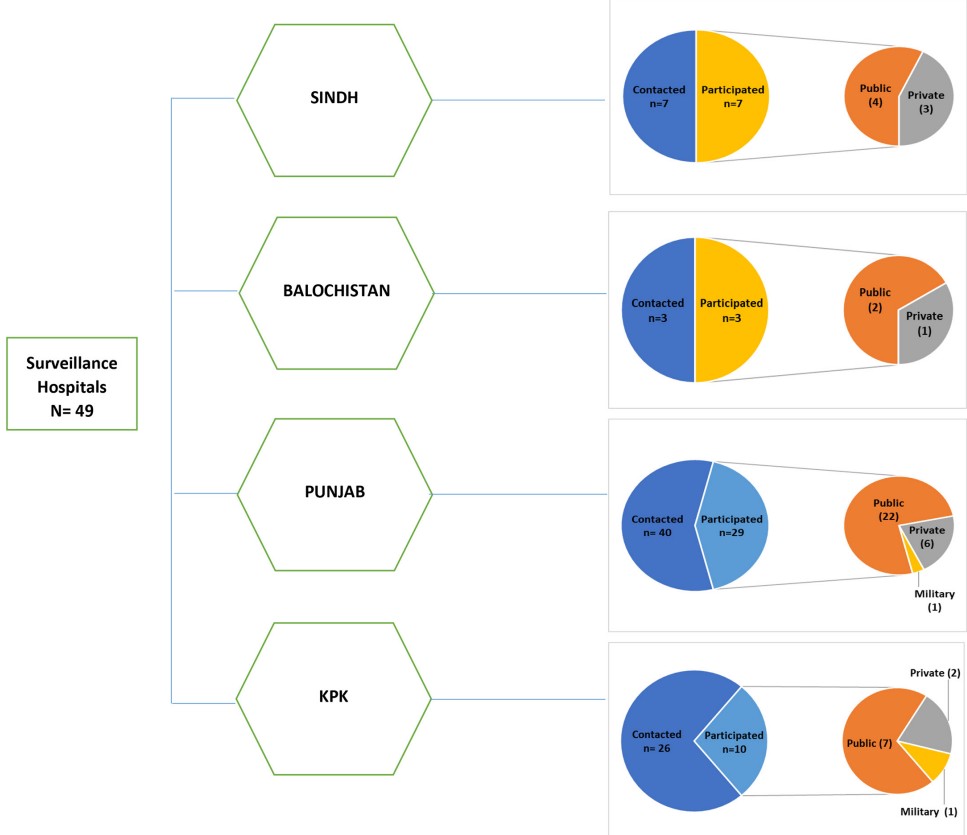

**Figure 1** Participatory tertiary care hospitals. KPK, Khyber Pakhtunkhwa.

### Follow-up with the iVDPV cases

The parents or guardians of iVDPV positive children are informed and followed up closely by Expanded Programme on Immunization (EPI) and Union Council teams along with WHO provincial teams. Children are followed up until two consecutive negative stool samples for a period of 6 months. However, if the child is still found to be an excretors at 6 months, they will be provided antiviral therapy for poliovirus excretion. This therapy would be provided to the affected child free of cost.

The children who test negative for VDPVs or Sabin-like virus, follow-up stool sample collection will be done after a year.

### Process

In order to maximise the surveillance coverage of primary immune deficient children, hospital paediatricians will be sufficiently sensitised to the necessary referral process.

Children with suspected PID screened using JMF warning signs will be enrolled and the following laboratory investigations would be sent for confirmation (figure 2).

► CBC (absolute neutrophil and lymphocyte count).
► HIV.
► Flow cytometry (B and T cell markers).
► QIGs test.

### Patient and public involvement

Considering the nature of the study, it was not possible to have a priori patient involvement. The country leadership from WHO and National Emergency Operations Centre were consulted during the study designing to safeguard public interest and safety.

### Biological sample collection, transport and laboratory processing

#### Blood samples

##### HIV

For HIV testing, 3–5 mL of whole blood will be collected in a gel tube and K2-EDTA tube. Gel tube will be used for serology whereas, EDTA tube would use for PCR. Samples will be transported to the laboratory at 2°C–8°C. Initial HIV screening will be performed by chemiluminescent immunoassay using ADVIA Centaur HIV Ag/Ab Combo assay kit run on ADVIA Centaur XP/CP analyser. Positive cases will be further confirmed by immunochromatographic antibody detection on Bio-Rad Genius HIV 1/2 assay. In case of indeterminate results or discrepant results between the above two methods final confirmation will be made by HIV PCR on Xpert HIV-1 Qual Assay kit and GeneXpert Dx System.

##### Complete blood count

For CBC, 3–4 mL whole blood sample will be collected in tubes containing K2 EDTA as an anticoagulant. Samples

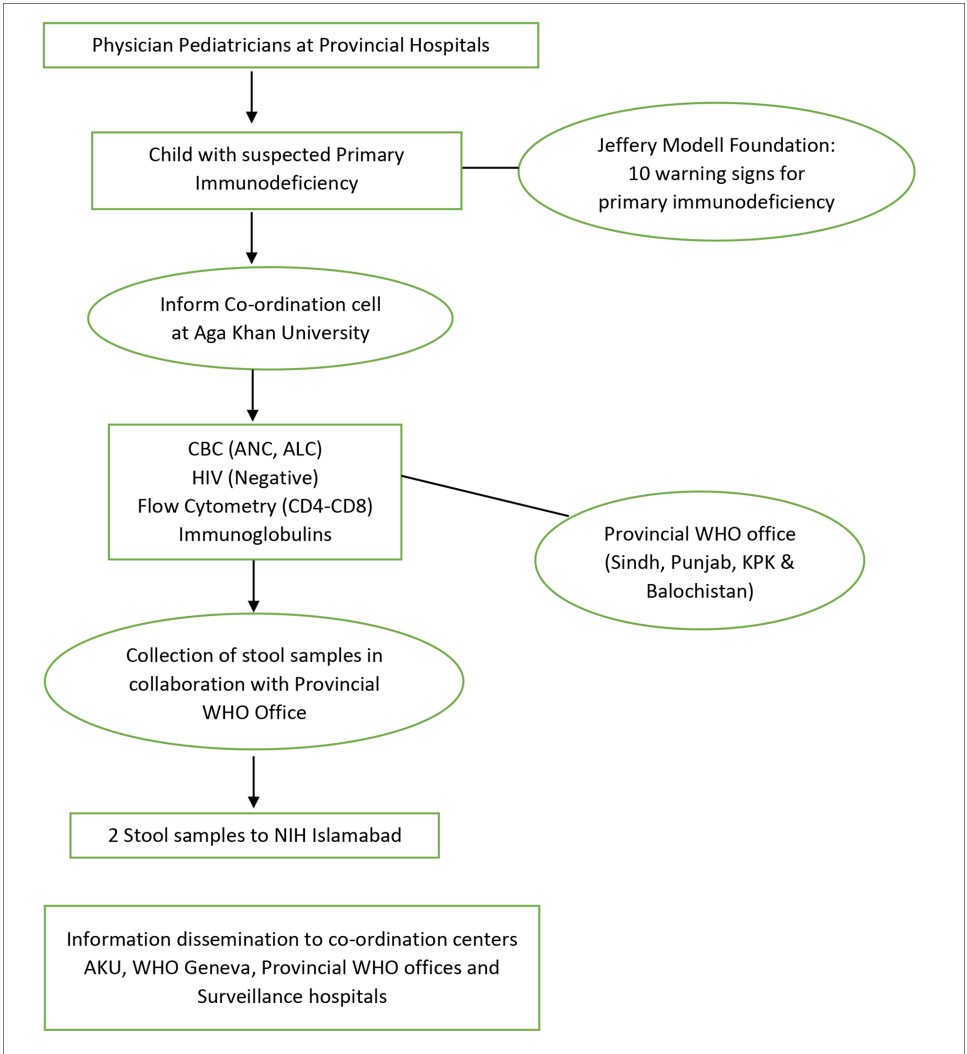

**Figure 2** Flow diagram of the study process. AKU, Aga Khan University; CBC, complete blood count; ANC, Absolute Neutrophil Count; ALC, Absolute Lymphocyte Count; KPK, Khyber Pakhtunkhwa; NIH, National Institute of Health.

will be transported as soon as possible to the stat/main laboratory. The temperature of 18°C–25°C will be maintained if the expected delivery time is within 8 hours, whereas 2°C–8°C will be maintained if samples are delivered after 8 hours up to a maximum of 3 days. Samples will be run on the fully automated Sysmex XN series analysers (Sysmex Japan).

### Immunoglobulins

For immunoglobulin measurement, 3–5 mL of whole blood will be collected in a plain tube or gel tube. Transportation temperature of 18°C–25°C (for 8 hours) and 2°C–8°C after 8 hours to 24 hours will be maintained. Serum IgA, IgG, IgM will be measured on ADVIA 1800 Chemistry analyser (Siemens Healthcare Diagnostics, New York, USA).

### Flow cytometry

For lymphocyte subset analysis, 3–4 mL of whole blood will be collected in tubes containing K2 EDTA or sodium heparin as an anticoagulant. All samples will be transported at 2°C–8°C within 24 hours. Lymphocyte subset

analysis will be performed by flow cytometry using FC500 MCL Flow Cytometer (Beckman Coulter, Miami, Florida, USA). Monoclonal antibodies against following antigens were used: T-lineage-associated antigens CD3, CD4 and CD8; B-lineage antigen CD19 and Natural Killer cell antigen CD56.

For CBC and immunoglobulin levels, three different levels of manufacturer-provided controls (low, normal and high) will be run with each batch as an internal quality control measure. Whereas, for lymphocyte subset analysis two levels (low and normal) commercial controls will be used. External proficiency is assured by simultaneously analysing samples from the College of American Pathologists two times per year for immunoglobulin levels and three times per year for both CBCs and lymphocyte subset analysis.

Following samples will be considered unsatisfactory for analysis: unlabeled or mislabeled samples, identification mismatch between specimen and requisition slip, haemolysed or clotted samples, wrong collection tube, insufficient specimen quantity, excessive delay in specimen

transport or improperly transported (ie, not at required temperature).

### Stool sample

Testing of stools for the isolation of poliovirus would be done according to the standardised protocol used at the Global Polio Laboratory Network.[8 9] Two stool specimens from the enrolled suspected primary immunodeficient children will be collected in sufficient quantity (~10 g) 24 hours apart. Then the samples will be shipped to WHO Regional Reference Laboratory for Polio Eradication Initiative, Virology Department, NIH-Islamabad, Pakistan within 48 hours maintaining reverse cold chain along with patient's investigation questionnaire (also known as lab request form). The lab request form indicated all general information such as demographic details (patient's name, age, sex and region), vaccination history and clinical information. Each sample will be provided a unique identity code (specimen number).

Samples will then be tested for isolation of specific serotypes. Stool sample processing will start with the tissue culture in which specimen will be grown. Second, the molecular method will be used for the isolation and identification of poliovirus serotypes. Third, two different assays of one-step real-time reverse transcriptase PCR (rRT-PCR) will be used for detection of polioviruses using the protocols and primers/probes corresponding to 5' UTR and VP1 region. These assays will include Intra-typic differentiation rRT-PCR that is for the confirmation, serotype identification and characterisation for polioviruses. However, rRT-PCR will be used for VDPV screening for VDPVs screening. Testing will be done according to the WHO guidelines for the identification of poliovirus.

### Data management and quality assurance
#### Data monitoring

Regular supervision for the identification of PID children at the selected surveillance hospital all over Pakistan will be done by the iVDPV Working Group members (WGM). From each hospital, there would be two designated members who would be comprehensively trained on Jeffrey Modell Signs and identification of primary immunodeficiency by the study PI. These members will be held responsible for correct identification of JMF signs, reviewing the filled surveillance questionnaire to ensure that it is filled properly and training the postgraduate students, residents and fellows on fundamentals of the project. Moreover, the project PI and RC of the study will also review forms and make visits to monitor the surveillance activity and to guide the iVDPV-WGM whenever required. This iVDPV-WGM will also update RC about the daily progress of the study via phone calls or by sending a medical history summary report of suspected PIDs on the WhatsApp group generated for this study.

Quality of data collection will be ensured by proper training of the iVDPV-WGM. Quality will also be ensured through supervision and reviewing of the data collected by the PI. Moreover, skype/zoom meetings will be arranged to maintain coordination among the members of the team and to keep the members up-to-date and to resolve any coordination-related issues.

#### Data editing

Data editing will be done in two phases: field site editing and office-based editing. Field editing will be done by the designated member of the surveillance site where they will check each questionnaire filled by attending physician on the hard copy of the questionnaire at the time of enrolment. These members will send the questionnaire to study RC for quality control and would be responsible for reviewing/editing if required at the central office-Aga Khan University, Karachi-Pakistan campus.

#### Data entry and cleaning

Two data entry operators will enter the data into a data entry application in parallel. Data entered will be compared for entry errors and the identified errors will be corrected. The hard copies of surveillance questionnaire will be kept at the secure location until the project is being completed.

### Plan of analysis and sample size

Data will be analysed using STATA V.11. Descriptive data analysis will be performed by calculating frequency with percentages for categorical data and mean with SD or median with IQRs for continuous scale data. We will calculate hospital-based proportions of total iVDPV cases in 2-year period. For sensitivity and specificity of the JMF signs, we will calculate the receiver operating characteristics curve by comparing the results based on screening with JMF signs with the results of flow cytometry and QIG tests. Since this is a surveillance, sample size calculation in not obligatory. However, for the sensitivity and specificity of JMF signs a minimum sample size of 50 PID cases, with 5% level of significance, change in sensitivity from 0.5 to 0.7 using two-sided binomial test and 78% power to detect a change in specificity from 0.5 to 0.7 using two-sided binomial test.[7]

### Limitations and strategies to overcome

As PID is a rare condition and many children die before formal diagnosis, we may not be able to achieve the required sample size of 50 PID cases in 2-year duration. While the minimum sample size is not a requirement for the surveillance objective, it is needed for the calculation of sensitivity and specificity of the JMF 10 warning signs. To achieve sufficient sample size, the PI of the project will review the enrolment data quarterly and may decide to expand the sentinel sites to other tertiary care hospitals centres in the country to increase the number of enrolments.

### Ethics and dissemination
#### Ethical approval

The surveillance project has been initiated after the formal ethical approval granted by the WHO, Aga Khan

University, National Bioethics Committee and Departmental Heads of Paediatrics from each participatory hospital.

## Informed consent

The formal written consent is taken from the parents or guardians of children with PID before the enrolment for blood and stool samples. Study details and processes are explained in detail, thumb impression is taken instead of a signature for parents who would be unable to read and write. Furthermore, it is recognised that no additional risk is involved as the blood collection for diagnostic testing is standard of care, and the stool collection poses no risks.

For HIV testing, we offer pre and post-test counselling to the families. If the test comes out positive, a referral will be made to the nearest National Aids Control programme for free treatment and follow-ups.

## Privacy and confidentiality

The information from the participants will be taken by maintaining privacy and confidentiality. A unique ID will be issued to each participant based on the province and hospital so that the names of any participant might not be disclosed. For maintaining participant's confidentiality, all data forms, reports, and other records will be identified by the same unique identifier. All records will be kept in a secure location and will be shared only with those who will be directly involved in the study such as PI and the coordination team members at AKU and WHO. All the computerised entries will be done with unique ID's only.

## Timeframe

This is a 2-year project to commence after approval of Ethical Review Committee (ERC).

## Dissemination plan

The scientific paper resulting from this work will be submitted for the publication to peer-reviewed journal. This is the main study protocol paper which serves to inform about the design, methodology and coordination of the project. Once the surveillance is completed, study results will be published as a separate scientific paper. Abstracts would be submitted at relevant conferences, if selected will be presented in the form of poster or presentation. Study participants will be informed about the stool sample results as soon as it is available and shared by the NIH. In case of positive results for VDPV, parents will be told about further steps. A dissemination seminar will be conducted for the members of iVDPV working group, WHO, policymakers, paediatric consultants and fellow researchers with same domain interest.

## Way forward

Pakistan National Registry of Primary Immunodeficiency disease will be developed. The aim of the national PID registry will be to establish the epidemiological data of primary immunodeficiency. Moreover, the registry would serve as a database for the coordination cells on PID in Pakistan.

**Acknowledgements** The authors would like to extend their gratitude to Dr Priya Sukhtankar for English proofreading and editing. She is affiliated with Department of Paediatrics, Gloucestershire Royal Hospital NHS Foundation Trust.

**Contributors** AFS has developed the research question, conceptualised the study and led the group that secured funding. MO has an advisory role for study conceptualisation and execution. MTY advised on methods, sample size calculations and statistical analysis. UN and MSR helped in liaison and support for iVDPV Surveillance in Pakistan. ASP and ZK are responsible for study management and coordination and drafted the manuscript along with AFS and MO. All authors have read, commented on and approved the final manuscript.

**Funding** This work is supported by WHO, grant number [2018/811124-0].

**Competing interests** None declared.

**Patient and public involvement** Patients and/or the public were not involved in the design, or conduct, or reporting, or dissemination plans of this research.

**Patient consent for publication** Not required.

**Provenance and peer review** Not commissioned; externally peer reviewed.

**ORCID iDs**
Asma Sadruddin Pethani http://orcid.org/0000-0001-9157-1711
Ali Faisal Saleem http://orcid.org/0000-0003-1804-9868

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
