## [Reviewer comments · BMJ Open]

ARTICLE DETAILS

TITLE (PROVISIONAL)	Poliovirus Excretion Among Children with Primary Immune Deficiency in Pakistan- A Pilot Surveillance Study Protocol
AUTHORS	Pethani, Asma; Kazi, Zaubina; Nayyar, Ujala; Shafiq-ur-Rehman, Muhammad; Yousafzai, Muhammad Tahir; Ondrej, Mach; Saleem, Ali

VERSION 1 – REVIEW

REVIEWER	Gamage, Deepa Ministry of Health, Sri Lanka, Epidemiology Unit
REVIEW RETURNED	21-Dec-2020

GENERAL COMMENTS	1. Good article and aim. Research question addressing is a requirement for Pakistan at this stage of Polio Eradication Programme and also as a part of VDPV prevention. 2. Suggest to maintain the uniformity of the wording used for the tool throughout the article as you start as "Jeffrey Modell warning signs" (JMF signs) of PID. please see line 105 JMF tool and also other places 2. using the JMF signs as a tool for screening to detect PID for iVDPV detection and following up is the main focus of the study and clearly explained. But, better explain that early detection of sabin like polioviruses excretion and following up for early detection of iVDPV and prevention of the transmission and outbreaks of VDPV as a requirement
--

REVIEWER	Khayeka-Wandabwa, Christopher African Population and Health Research Center (APHRC), Nairobi, Kenya
REVIEW RETURNED	28-Feb-2021

GENERAL COMMENTS	The protocol focuses on a study that seeks to establish a pilot surveillance system for successful identification and monitoring of vaccine-derived poliovirus excretion among children with primary immunodeficiency disorders (PID). Implementation of the protocol is meant to provide baseline evidence on whether Jeffrey Modell warning signs of PID can be utilized as an appropriate screening tool for PID in Pakistan. The protocol is well described with clear work flow for each data points to be examined. Equally, the study is meant to relay evidence on a worthwhile course of critical importance to many developing nations. I have two clarification questions: -Page 5 of 15; Line 86-87 what was the basis of identifying the participatory tertiary healthcare facilities included in this study? -Page 10 of 15; Line 250 sample size: With the understanding that PID is a rare condition and many children die before formal
---

	diagnosis, what would be your approach in calculating sensitivity and specificity of the JMF 10 warning signs if sufficient sample size of 50 PID cases is not achieved even after your intervention plan of expanding the sentinel sites to other tertiary care hospitals centers?
--	---

VERSION 1 – AUTHOR RESPONSE

S. No.	Reviewer's Comments	Responses
Reviewer: 1		
1	Good article and aim. Research question addressing is a requirement for Pakistan at this stage of Polio Eradication Programme and also as a part of VDPV prevention.	Acknowledged.
2	Suggest to maintain the uniformity of the wording used for the tool throughout the article as you start as "Jeffrey Modell warning signs" (JMF signs) of PID. please see line 105 JMF tool and also other places.	The word tool has been revised as a screening tool. (Line No.: 20, 75, 83) However, JMF tool has been revised a JMF warning signs. (Line No.: 107)
3	Using the JMF signs as a tool for screening to detect PID for iVDPV detection and following up is the main focus of the study and clearly explained. But, better explain that early detection of sabin like polioviruses excretion and following up for early detection of iVDPV and prevention of the transmission and outbreaks of VDPV as a requirement.	It is a very worthy point; we have added this important comment in line no. 80-83.
Reviewer: 2		
4	Page 5 of 15; Line 86-87 what was the basis of identifying the	Most of the children with suspected or confirmed PID end up reporting to public/private tertiary care hospitals. Almost all the tertiary care hospitals have well established clinical

	participatory tertiary healthcare facilities included in this study?	paediatric faculty with strong academic, clinical and administrative units. Hence, this is vital to involve tertiary care hospitals. While efforts were made to involve all public sector tertiary care hospitals, however, due to administrative/ managerial issues could not be present at the master training session and therefore were not included as a participatory hospital. However, as the study progresses efforts are underway to induct more tertiary care hospitals. The network of Acute Flaccid Paralysis which includes; national surveillance coordinator, WHO regional reference laboratory coordinator, federal and provincial surveillance officers have a good reputation within these tertiary care facilities.
5	Page 10 of 15; Line 250 sample size: With the understanding that PID is a rare condition and many children die before formal diagnosis, what would be your approach in calculating sensitivity and specificity of the JMF 10 warning signs if sufficient sample size of 50 PID cases is not achieved even after your intervention plan of expanding the sentinel sites to other tertiary care hospitals centers?	It is an important comment. The authors also have the similar opinion regarding rare disease of PID in community. However, in Pakistan half of the population marry a first or second cousin [Reference: Pakistan Demographic and Health Survey 2017-18]. With the understanding that PID is almost all the time inherited disorder, the sample size was inflated to 50 with this assumption. As the surveillance is already in progress, we have been able to enroll 172 children using JMF warning signs. More than 60% have confirmed PID. (Analysis in process)